# The Relationship between Reticuloruminal Temperature, Reticuloruminal pH, Cow Activity, and Clinical Mastitis in Dairy Cows

**DOI:** 10.3390/ani13132134

**Published:** 2023-06-28

**Authors:** Ramūnas Antanaitis, Lina Anskienė, Giedrius Palubinskas, Arūnas Rutkauskas, Walter Baumgartner

**Affiliations:** 1Large Animal Clinic, Veterinary Academy, Lithuanian University of Health Sciences, Tilžės Str. 18, LT-47181 Kaunas, Lithuania; 2Department of Animal Breeding, Veterinary Academy, Lithuanian University of Health Sciences, Tilžės Str. 18, LT-47181 Kaunas, Lithuania; 3Clinic for Ruminants, University of Veterinary Medicine, Veterinaerplatz 1, A-1210 Vienna, Austria

**Keywords:** precision farming, parameters, diseases, dairy cows

## Abstract

**Simple Summary:**

Automated monitoring devices are becoming more widespread in the dairy sector. Upon a literature analysis, we found that more research is needed to assess mastitis status, including spontaneous infections with various pathogens and other variations in reticulorumen temperature. Based on the literature review, we aimed to determine the relationship between reticuloruminal temperature, pH, cow activity, and clinical mastitis (CM) in dairy cows. According to our study findings, we concluded that reticuloruminal temperature, reticuloruminal pH, and cow activity could be used as parameters for the early diagnosis of CM in dairy cows. Furthermore, CM occurrences must be tracked over shorter periods so that farmers can receive information to make wise management decisions.

**Abstract:**

We hypothesized that reticuloruminal temperature, pH as well as cow activity can be used as parameters for the early diagnosis of clinical mastitis in dairy cows. Therefore, we aimed to detect the relationship between these factors and the disease. We randomly selected cows with clinical mastitis and clinically healthy cows (HG) out of 600 milking cows. We recorded the following parameters during the experiment: reticulorumen temperature (RR temp.), reticulorumen pH (RR pH), and cow activity. We used smaXtec boluses (smaXtec animal care technology^®^, Graz, Austria). In this investigation, reticulorumen data obtained seven days before diagnosis were compared to HG data from the same time period. CM cows were observed on the same days as the healthy cows. The healthy group’s RR pH was 7.32% higher than that of cows with CM. Reticulorumen temperature was also 1.25% higher in the CM group than in the control group. The healthy group had a higher average value for walking activity, which was 17.37% higher than the CM group. The data of reticulorumen pH changes during 24 h showed that during the day, the pH changed from 5.53 to 5.83 in the CM group. By contrast, pH changed from 6.05 to 6.31 in the control group. The lowest reticulorumen pH in the CM group was detected on the third day before diagnosis, which was 15.76% lower than the highest reticulorumen pH detected on the sixth day before diagnosis. The lowest reticulorumen pH in CM cows was detected at 0 and 1 days before diagnosis and it was 1.45% lower than the highest reticulorumen pH detected on the second day before diagnosis. The lowest walking activity in the CM group was detected 0 days before diagnosis, which was 50.60% lower than on the fifth day before diagnosis. Overall, the results confirmed our hypothesis that reticuloruminal temperature, reticuloruminal pH, and cow activity could be used as parameters for the early diagnosis of clinical mastitis in dairy cows.

## 1. Introduction

Bovine mastitis, caused by complex interactions between the host, environment, and infectious organisms, is one of the most common diseases of dairy cattle. It significantly impacts global dairy production by reducing milk quantity and quality [1]. Aside from its direct effects on dairy production, bovine mastitis is also a serious human health concern as it contains a major infectious agent called *Staphylococcus (S.) aureus* [2]. *S. aureus* is a well-known bacterium whose strains with high pathogenic potential (e.g., harboring virulence factors and antimicrobial resistance genes) can infect both animals and humans through a variety of channels, including food. It is also one of the primary causes of bovine mastitis [3]. Food-producing animals, particularly cattle and pigs, are significant producers of *S. aureus* in the food chain [4]. Mastitis can be characterized as infectious or environmental, depending on the micro-organisms that cause the infection. During milking, contagious bacteria spread across infected quarters [5]. *S. aureus* is a facultative anaerobe with capsule, nonmotile, and non-spore-forming cocci that are Gram-positive, catalase- and coagulase-positive. This bacterium is regarded as one of the primary pathogens involved in bovine mastitis worldwide because it is highly contagious and causes long-term chronic infections [6]. Although several mastitis prevention initiatives have been adopted in various countries, *S. aureus* prevalence in cows remains high [7]. Mastitis causes an inflammatory reaction which increases the somatic cell count, which then lowers synthetic activity in the mammary tissue and alters milk composition. Furthermore, increased vascular permeability can result in ions, protein, and enzyme leakage. It also lowers milk quality by decreasing the production of various milk components, such as lactose, fat, non-fat solids, and casein. Some negative impacts of mastitis include milk production losses, medicine and veterinary service expenditures, and death or early euthanasia of affected animals [8]. Aside from its negative impact in animal production, *S. aureus* in milk and other dairy products can cause human infections, which are important public health concerns. Staphylococcus aureus is a common healthcare-acquired infection that has recently become a community-acquired pathogen [9]. *S. aureus* strains isolated in humans can swap hosts and adapt to cows. Monitoring *S. aureus* lineages is important to limit hazard spread in both the herd and the population [10]. These infections can be more severe if the microbe manufactures toxins [11]. Effective diagnostic approaches can facilitate mastitis control and encourage antimicrobial stewardship. Clinical mastitis must be objectively scored to predict therapy outcomes and adjust treatment procedures accordingly [12]. 

Recently, in the dairy sector, automated monitoring devices (AMDs) have become more widely available [13]. Adoption of AMDs is likely to increase as dairy herd size and consolidation continue [14,15,16]. Since clinical mastitis (CM) is a severe concern for dairy farmers, many AMDs have been developed to aid in early diagnosis [17,18]. Given the difficulties in visually analyzing milk quality in herds with automatic milking systems, applying AMDs to detect intramammary infection (IMI) can be critical [1]. Milk yield, composition, somatic cell count (SCC), electrical conductivity, and flow rate are commonly used to determine IMI [19]. Furthermore, activity-recording devices that count steps or time spent lying, standing, walking, or ruminating have been used with different degrees of efficacy [20,21]. Continuous observation of reticuloruminal pH data has revealed a robust and predictable short-term pattern that can be characterized by a simple sine wave and frequency of one cycle per day [22]. Thus, evaluating reticuloruminal pH is of interest to most doctors, who define the state based on single observations. Continuous monitoring of pH values using remote sensing data is now possible; however, this generates a massive volume of data that can be difficult to comprehend. As a result, researchers who use continuous pH monitoring techniques to study food and acidosis frequently use average values or threshold procedures to evaluate reticuloruminal pH [22].

Because of technological advancements and growing interest in AMDs, continuous measurement of reticular pH is now possible using indwelling devices (boluses) [23]. 

Previous studies showed that body site temperature is an effective technique for determining IMI due to the feverish state caused by the immune response to infections [24,25]. However, most studies have concentrated on the udder skin surface or rectal temperature, with the latter being the most common approach on dairy farms [20,26]. Intra-reticuloruminal AMD provides an alternative to physically monitoring rectal temperature, allowing for automatic and regular monitoring without holding the cows [27]. The period when rumen pH is below a given threshold is traditionally used to explain variation in rumen pH [28], albeit threshold levels vary between research and monitoring methods [29]. A bolus placed into the rumen can measure a cow’s temperature in real time (reticulorumen). Temperature and pH can be measured using boluses. Wireless boluses send data every ten minutes. The information can be saved in the cloud or on a computer. Measurements can be recorded for up to a year, depending on the battery life of the various bolus variants [30]. More research is needed to assess mastitis status, including spontaneous infections with various pathogens and other variations in reticulorumen temperature [31]. In our past studies, we determined that veterinarians and farmers should examine the likelihood of stillbirth during late gestation based on clinical mastitis. This practice may help develop methods for enhancing reproductive performance in dairy cows [32]. The findings show that clinical mastitis impacts the time and rumination chews registered by sensor systems [33]. 

For this study, we hypothesized that reticuloruminal temperature, reticuloruminal pH, and cow activity could be used as parameters for the early diagnosis of clinical mastitis in dairy cows and thus aimed to determine the relationship between these factors and the disease. 

## 2. Materials and Methods

### 2.1. Ethical Approval

The Lithuanian Law on Animal Welfare and Protection was followed in conducting this investigation. The approval number of this study is G2-227. The State Food and Veterinary Service’s Department of Animal Welfare provided the ethical approval post an in-person review of the experimental setup. 

### 2.2. Animals Farm and Feeding 

From July to December 2022, the experiment was conducted at the Lithuanian University of Health Sciences and a dairy farm (55.792368° N, 24.017499° E) with 600 milking Holstein cows. The study was conducted with cows at a second lactation, with an average daily milk yield of 35 kg per cow, an average feed intake of 19 kg dry matter (DM)/day, milk fat of 4.3, milk protein of 3.55, an average of milk somatic cell count of 190,000/mL, and an average of milk urea nitrogen of 22%. Cows were milked using a DeLaval milking parlor twice daily and kept in a stall-free barn.

Individual attribute data (lactation number, breed, latest calving date, and milk yield) were acquired from the farm’s computer system and documented on a spreadsheet (Delpro DeLaval Inc., Tumba, Sweden). The number of days in milk (DIM) for each cow was obtained for each data collection period by calculating the number of days between the last calving date and the first day of the data collecting period. All cows were fed a total mixed ration (TMR) consisting primarily of maize and alfalfa silage. TMR comprised 38% grass silage, 38% corn silage, and 24% flaked grain concentrate with a mineral mixture. Depending on the circumstances, the ration was designed to meet or exceed the needs of a 550 kg Holstein cow producing 35 kg of milk daily. Cows were fed daily at 8:00 a.m. and 4:00 p.m.

### 2.3. Research Design

From 1 July to 31 December 2022, 28 clinical mastitis cases from herd of 600 cows were identified by the farm staff in the milking parlor and confirmed by a local veterinarian based on the clinical indications/sigs of the mastitis, such as abnormal milk appearance (watery, flakes, fibrin clots, etc.; mild CM), abnormal milk appearance with a swollen or painful quarter (moderate CM), abnormal milk appearance with a swollen or painful quarter, and systemic signs of illness (fever, decreased appetite, dehydration, etc.; severe). Cows with mild and moderate CM were included in the study. In this research, we included cows with all four infected quarters. This group comprised 28 cows (187.5 ± 5 days in milk).

A general clinical examination revealed that no cows had clinical indications compatible with disease or other variables, such as heat stress or estrus. Cows with these characteristics (n = 3) were excluded from the research. Cows classified as control did not have mastitis during their current lactation. This group comprised 25 cows (195.65 ± 2.5 days in milk). If the cow met the enrollment criteria, local veterinarian aseptically collected a single milk sample from the affected quarter.

All cows’ (n = 25) milk samples were collected for microbiological investigation. Using the techniques indicated in the National Mastitis Council guidelines [34], and detailed laboratory protocols reported elsewhere [35], 20 cases of CM caused by *S. aureus* and 5 caused by *Streptococcus* spp. were detected during the study.

The antimicrobial susceptibility of *S. aureus* isolates was tested using a broth microdilution panel incorporating antimicrobials used in the treatment of mastitis. The tests were carried out according to the manufacturer’s instructions [36]. The microbiology test showed *Streptococcus* spp. and *S. aureus*, as well as amoxicillin and clavulanic acid sensitivity. Cows with CM were treated with intramammary antibiotics and anti-inflammatory drugs (Synulox LC + NSAID). The latter included Melovem^®^ 20 mg/mL. Cows were treated with a single subcutaneous injection at a dosage of 2.5 mL/100 kg of body weight. The antimicrobial used was Synulox LC (for lactating cows). Each 3 g syringe contained 200 mg of amoxicillin (as amoxicillin trihydrate), 50 mg of clavulanic acid (as potassium clavulanate), and 10 mg of prednisolone that was administered by intramammary infusion soon after milking and at 12 h intervals for three consecutive milkings. Following the final milking, antibiotic infusions were administered as follows: trained staff wearing clean disposable gloves cleansed the teat ends for at least 5 s with 70% isopropyl alcohol-soaked cotton swabs before the antibiotic treatment was infused into the mammary gland [37]. 

Twenty-five cows (second and more lactation numbers and 193.65 ± 2.4 days in milk) were categorized as clinically healthy with an SCC of <200,000 cells/mL and no clinical signs of disease, such as abnormal milk appearance (watery, flakes, fibrin clots, etc.; mild CM), abnormal milk appearance with a swollen or painful quarter (moderate CM), abnormal milk appearance with a swollen or painful quarter, and systemic signs of illness (fever, decreased appetite, dehydration, etc.; severe).

### 2.4. Measurements

#### 2.4.1. Measurement Equipment

The reticulorumen parameters and walking activity of all cows (n = 600) were assessed using smaXtec boluses (smaXtec animal care technology^®^, Graz, Austria), designed specifically for animal care. SmaXtec animal care technology^®^ allows for the continuous presentation of data, such as ruminal pH and temperature, in real time. The boluses were injected into the reticulorumens using a special tool, as directed by the manufacturer. One bolus is delivered orally per cow and is retained in the reticulum due to gravity. The boluses were activated before application, cross-referenced with the cow’s unique ear tag number, and a connection to the base station was established. The device’s proper operation was checked, and it was calibrated in a buffer solution with a pH of 7.00 (Reagecon Buffer Solution 10702550, Reagecon, Shannon, Ireland). Following a general examination of cows in a self-locking grid, heads were manually restrained by the individual applying the bolus. The mouths were opened and boluses were applied with the proper applicator (1.34 inch balling gun). Boluses were transported to the base of the tongue and ingested voluntarily by cows. Cows were then watched for two hours for any undesirable effects. 

The data were collected using antennas (smaXtec animal care technology^®^). An indwelling and wireless data-transmitting device was used to monitor RT, pH, TRR, and cow activity (smaXtec animal care GmbH, Graz, Austria). A microprocessor controlled the system. Data on pH and TRR were acquired using an A/D converter and stored on an external memory chip for further analysis and interpretation. The smaXtec messenger^®^ computer software (version 4) collected all data (smaXtec animal care technology^®^, Graz, Austria).

We recorded the following parameters during the experiment: reticulorumen pH (pH), reticulorumen temperature (RR temp.), and cow activity. Using smaXtec boluses, we could monitor real-time data, such as pH, reticulorumen content (TRR) temperature, and cow activity [38,39]. 

#### 2.4.2. Duration of Measurements 

In this investigation, we compared reticulorumen data from a week before diagnosis with HG data from the same time period. CM cows were observed on the same days as healthy cows.

#### 2.4.3. Statistical Analysis 

The experimental animals were divided into four classes based on the reticulorumen pH assay: first class < 6.22; second class 6.22–6.42; third class 6.42–6.62; and fourth class > 6.62. Classes were assigned according to our previous publication [40]. SPSS 25.0 (SPSS Inc., Chicago, IL, USA) was used for statistical data analysis. Using descriptive statistics, the Kolmogorov–Smirnov test assessed the normal distribution of variables. A linear regression equation was calculated to determine the statistical relationship between the study’s recorded parameters—reticulorumen pH (pH), reticulorumen temperature (RR temp.), and cow activity (dependent variables)—in the days before diagnosis or between reticulorumen pH classes (independent indicators). A backward stepwise logistic model was applied to exclude all non-essential explanatory variables (according to the Wald test’s significance). The estimates and 95% Wald limits were used to calculate the odds ratio (OR) for the probability of success to failure and the 95% confidence interval (CI), which indicates a 95% probability that the true OR is likely to be within the specified range. The Pearson correlation was used to detect the linear relationship between the investigated traits.

Repeated measures analysis of variance (ANOVA) was used for comparing means across the investigated variables based on observations from 7 to 0 days before diagnosis. 

## 3. Results

The distribution of cows according to reticulorumen pH classes showed 84.62% more cows in the first class of reticulorumen pH than in the fourth class (χ^2^ = 7.111, df = 1, and *p* = 0.01). There were 61.54% more cows in the first class than in the second and third (χ^2^ = 16.133, df = 1, and *p* = 0.001) and 60.00% more cows in the second and third class of reticulorumen pH than in the fourth class (χ^2^ = 2.571, df = 1, and *p* > 0.05), (Figure 1).

The average reticulorumen temperature of the first-class cows was 0.74% higher than in the third class (*p* < 0.01) and 0.13–0.38% higher than in the second and fourth classes (*p* > 0.05). Data analysis of the differences in means of reticulorumen temperature revealed a statistically significant relationship between classes of reticulorumen pH. The reticulorumen temperature decreased by about 0.029 °C (y = −0.029x + 39.02; and R^2^ = 0.0857). The average walking activity of first-class cows was 10.85% higher than in the fourth class and 2.48–4.34% higher than in the second and third classes (*p* > 0.05). Data analysis of walking activity showed a statistically significant relationship between reticulorumen pH classes. Walking activity decreased by about 0.222 steps/min (y = −0.222x + 6.72; and R^2^ = 0.9157) (Figure 2).

Analysis of the investigated traits in a group of cows showed that the highest average pH value was estimated in the control group, which was 7.32% higher than in the investigated group of cows (*p* < 0.001). Reticulorumen temperature was 1.25% higher in the investigated group than in the control group (*p* < 0.05). For walking activity, the higher average value was determined in the control group, which was 17.37% higher than in the investigated group of cows (*p* < 0.001) (Table 1).

The mean of reticulorumen pH changes of every 24 h showed that pH changed from 5.53 to 5.83 in investigated group. In the control group, pH changed from 6.09 to 6.35 (Figure 3). 

The lowest reticulorumen pH in the investigated group was detected on the third day before diagnosis. It was 15.76% lower than the highest reticulorumen pH detected on the sixth day before diagnosis in the investigated group (*p* < 0.001). The lowest reticulorumen pH in the control group was detected at 0 and 1 days before diagnosis in the CM group; it was 1.45% lower compared to the highest reticulorumen pH detected on day 2 before diagnosis in the CM group (*p <* 0.001). 

Analysis of the reticulorumen pH in the CM group of cows showed that the highest average pH value was estimated in cows on the sixth day before diagnosis. According to comparisons of pH means between groups, we recorded statistically significant differences on the third day before diagnosis (14.63% lower in the investigated group than in the control, *p* < 0.001), the second day before diagnosis (14.89% lower in the investigated group than in the control group, *p* < 0.001), first day before diagnosis (12.49% lower in the investigated group than in the control group, *p* < 0.001) and on the diagnosis day (13.63% lower in the investigated group than in the control group, *p* < 0.001). 

Data analysis of reticulorumen pH in the investigated group of cows revealed a significant relationship between days before diagnosis. Reticulorumen pH decreased by approx. 0.164/day in the investigated groups of cows (y = −0.1642x + 6.4346; and R^2^ = 0.790). The control group remained almost the same (y = −0.0057x + 6.1707; and R^2^ = 0.1558) (Figure 4).

The lowest reticulorumen temperature in the investigated group was detected on the fifth day before diagnosis. It was 2.08% lower than in the highest reticulorumen temperature detected on the third day before diagnosis in the investigated group (*p* < 0.001).

According to the reticulorumen temperature means between groups, we estimated statistically significant differences between all days, except the sixth day before diagnosis.

The reticulorumen temperature mean difference ranged from 0.56% on the seventh day before diagnosis, higher in the control group (*p <* 0.001), to 2.60% on the third day before diagnosis (*p <* 0.001).

Data analysis of reticulorumen temperature showed a significant relationship between days before diagnosis. The reticulorumen temperature increased by approx. 0.12 °C/day (y = 0.1221x + 38.811; and R^2^ = 0.6748), whereas in the control group, reticulorumen temperature remained almost the same (y = 0.008x + 38.94; and R^2^ = 0.0079) (Figure 5).

The lowest walking activity in the investigated group was detected at 0 days before diagnosis, 50.60% lower than day 5 before diagnosis (*p* < 0.001). 

Regarding the walking activity means between groups, the highest statistically significant differences were estimated seven days before diagnosis (27.41% lower in the investigated group than in the control group, *p* < 0.001), the first day before diagnosis (26.67% lower in the investigated group than in the control group, *p* < 0.001), and the sixth day before diagnosis (19.02% lower in the investigated group than in the control group, *p* < 0.001). 

Data analysis of walking activity showed a significant relationship between days before diagnosis. Walking activity decreased by approx. 0.472 steps/min/day (y = −0.472x + 7.5974; and R^2^ = 0.8302) in the investigated group. In the control group, we detected a similar relationship. Walking activity decreased by approx. 0.4957 steps/min/day (y = −0.4957x + 8.8432; and R^2^ = 0.9087) (Figure 6). 

The days before diagnosis showed low significant negative relationship with walking activity in both groups of cows (*p* < 0.001). In the investigated group, days before diagnosis showed a low significant positive relationship between reticulorumen temperature and a negative relationship with reticulorumen pH (*p* < 0.001). While in the controlled group, days before diagnosis showed a low significant positive relationship between reticulorumen pH (*p* < 0.001), and negative with temperature (Table 2).

We used a logistic regression model to determine the associations between factors that contributed to the chances of getting mastitis. 

One of the dimensions had an outcome of interest with two categories (first group: cows with clinical mastitis, and second group: healthy cows), indicating that subclinical mastitis among the cows was more likely associated with increased odds of reticulorumen temperature (OR = 1.260) and less likely with walking activity (OR = 0.875, and *p* < 0.001), and reticulorumen pH (OR = 0.156, and *p* < 0.001) (Table 3). 

## 4. Discussion

Due to increased interest and acceptance of automatic (robotic) milking systems (AMS), reliable automatic diagnosis of mastitis is needed to reduce the inspection time required to identify cows with mastitis that require veterinary care [21]. Efficient mastitis identification allows farmers to implement early and sufficient treatment procedures, reduce antibiotic overuse, conserve animal health and welfare by minimizing pain and suffering, increase recovery rates, and maximize economic return [41]. Moreover, accurate diagnosis methods can facilitate mastitis control and promote wise antimicrobial use [42]. We hypothesized that reticuloruminal temperature, reticuloruminal pH, and cow activity could be indicators for early detection of clinical mastitis in dairy cows and thus aimed to find a link between these factors and the disease. CM prediction using milk-related variables (e.g., conductivity, SCS, lactate dehydrogenase, and milk yield), alone or in combination, was widely tested with varying results [19]. Reliable mastitis diagnosis via automated technologies allows us to implement early treatment programs, avoid antibiotic misuse, conserve cow health and welfare, avoid discomfort and pain, and boost recovery rate and farm economic viability [43]. 

RR pH in the healthy group was 7.32% higher than in CM cows. Reticulorumen temperature was 1.25% higher in the CM group than in the control group. The healthy group had a higher average value for walking activity and was 17.37% higher than the CM group. The data on reticulorumen pH changes over 24 h showed that pH changed from 5.53 to 5.83 in the CM group. In the control group, pH changed from 6.05 to 6.31. According to the existing literature, ruminal pH is the best indicator for subacute ruminal acidosis (SARA) risk due to rumen pH variations in dairy cows [44]. As a result, cows’ vulnerability to SARA varies [45]. Rumination and fermentation processes are inextricably linked. Thus, decreased rumination activity leads to decreased saliva buffering, raising the risk of SARA [46]. Clinically sick cows generally exhibited a low rumen pH and reduced feed intake. It is unknown if low pH is a cause or is an outcome of the condition [47]. Decreased feed intake is a common sing of sickness. For example, Van Winden et al. [48] discovered a decreased total DMI prior to diagnosing displaced abomasum. In the investigation by Huzzey et al. [49], cows at risk of metritis could also be identified by total DMI. Lukas et al. [50] used DMI changes as an indication of health status. Denwood et al. [22] proposed forecasting illness episodes by monitoring deviations from an expected pattern (of feed intake or rumen pH) as a future study emphasis. In our previous study, we found that higher reticulorumen pH and temperature, higher milk lactose levels, and lower concentrate consumption are all associated with cow reproduction success. On the other hand, changes in cows’ productivity and activity have also been reported as dependable predictors of cow reproduction success [51]. 

In our study, the lowest reticulorumen pH in the CM group was detected on the third day before diagnosis. It was 15.76% lower than the highest reticulorumen pH detected on the sixth day before diagnosis. Although the lowest reticulorumen pH in CM cows was detected at 0 and 1 days before diagnosis, it was 1.45% lower than the highest reticulorumen pH detected on the second day before diagnosis. 

The use of RRT to predict CM has received little attention [31]. In this study, we found that the lowest reticulorumen temperature in the CM group was detected on the fifth day before diagnosis. It was 2.08% lower than the highest reticulorumen temperature detected on the third day before CM diagnosis. Adams et al. [27] evaluated the association between RRT and naturally occurring mastitis using a novel intra-reticuloruminal device that monitored the reticular temperature at each milking. The RRT was within the normal temperature range of 38.0–39.4 °C before the challenge [52] and increased following the intramammary challenge, remaining constant until the end of the study period [31]. RRT increased by 2.4 °C in quarters experimentally infected with *Escherichia coli*, according to AlZahal et al. [44]. We found a slight increase in RRT, likely due to pathogenicity differences between *Strep. uberis* and *E. coli*, highlighting pathogen-specific aspects of temperature fluctuations after IMI [31]. Increased RRT may be related to the number of bacteria in the infected quarters since an association between bacterial concentration and body temperature was previously reported [21,22]. Several parameters, including ambient temperature, drinking water temperature, feed intake, and diet composition, have been shown to alter the strength of the connections [31]. Research differences may be attributed to the algorithms and temperature limits used to create alarms. In our investigation, we set the alarm threshold to one standard deviation from the baseline RRT temperature [31]. The increase in RRT following an intramammary challenge with *Strep. uberis* triggered warnings based on one standard deviation of the baseline temperature [31]. According to the literature, the reticulorumen temperature during the week preceding the challenge was within the normal temperature range of 38.0–39.4 °C. It increased after the intramammary challenge, remaining steady until the end of the study period [52].

The lowest walking activity in the CM group was detected 0 days before diagnosis, which was 50.60% lower than day 5 before diagnosis. The lowest walking activity was detected 0 days before diagnosis, which was −39.57% lower than day 7 before diagnosis. Mastitis is one of the most serious diseases in dairy cows for it causes much pain in the afflicted animals. Additionally, it has widely known for its negative impacts on dairy farms’ welfare and profitability [53]. Increased animal activity prior to disease discovery may be related to increased stress [54]. Cows spent less time sleeping, ruminating, and drinking when their udders were highly swollen and fever-ridden [55]. Our previous research has shown that circadian rhythm and seasonal reproductive cycles alter nursing cows’ reticulorumen temperature and pH [56]. Changes in activity and knowledge about temporal effects at least 10 days following a mastitis diagnosis contribute to our understanding of naturally occurring cases [57]. During illness, laying increases across animal species. Reduced activity is useful for energy conservation and a febrile response [58]. As a result, decreased laying behavior may be perceived as a response to an udder infection in both experimentally produced and natural cases of acute clinical mastitis [57]. Relevant behavioral health indicators are described in order to integrate machine learning algorithms and decision support systems to detect animal lameness, lethargy, pain, injury, and distress. The roadmap for technology adoption is also highlighted, as are the difficulties and next steps. The technology has the potential to result in more effective farm animal management, a more focused emphasis on sick animals, medical expense reductions, and fewer antibiotic use [59]. In an ideal world, as future PLF performance improves, data from the bolus system should be integrated with incoming information from other systems (e.g., robotic milking system, accelerometer, and weather data) that are used to monitor the respective herd and the health and productivity of the herd’s individuals [23].

## 5. Conclusions

In this study, we found that reticuloruminal temperature, reticuloruminal pH, and cow activity could be used as parameters for the early diagnosis of clinical mastitis in dairy cows. The RR pH of healthy cows was 7.32% higher than CM cows. Data on reticulorumen pH changes over 24 h showed that CM cows’ pH changed from 5.53 to 5.83, whereas in the healthy group, the pH changed from 6.05 to 6.31. Regarding walking activity, a higher average value was determined in the healthy group, which was 17.37% higher than the CM group. 

CM occurrences must be tracked over shorter time periods so that farmers can receive information that facilitates wise management decisions. In future CM research, more cows should be studied, and this dataset should be used to create prediction models through machine learning approaches. Retrospective clinical mastitis prediction systems that use machine learning could benefit from these data.

## Figures and Tables

**Figure 1 animals-13-02134-f001:**
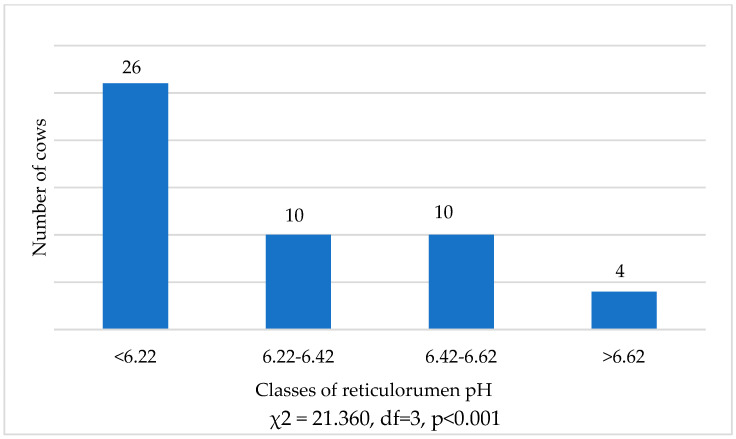
Distribution of cows (average pH during the research) according to reticulorumen class.

**Figure 2 animals-13-02134-f002:**
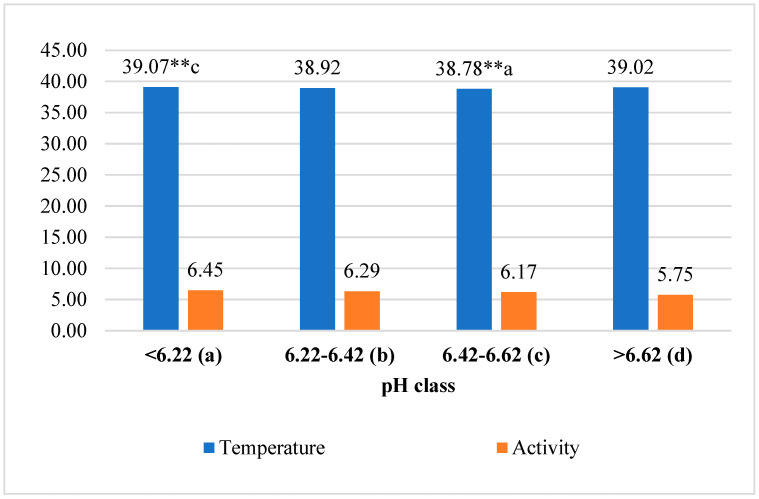
Analysis of investigated traits by reticulorumen pH classes. Different letters a, b, c, and d indicate statistically significant differences between means of different pH classes ** *p* < 0.01.

**Figure 3 animals-13-02134-f003:**
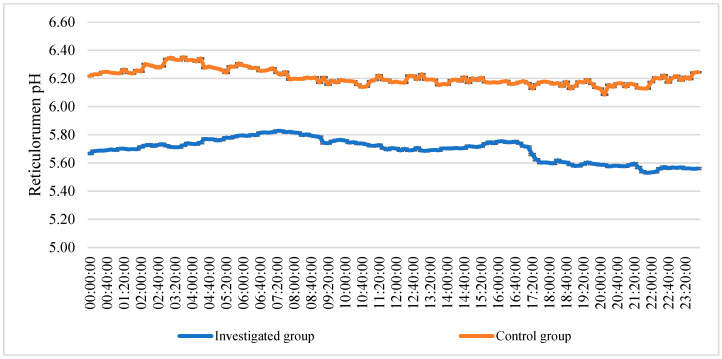
The mean of reticulorumen pH changes over 24 h in a group of cows.

**Figure 4 animals-13-02134-f004:**
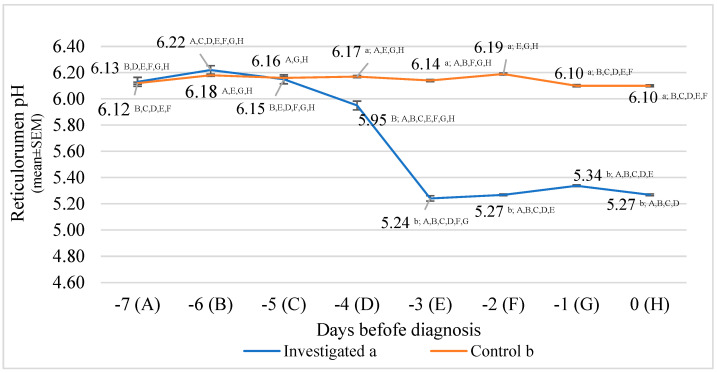
Comparison of reticulorumen pH between two groups of cow’s days before diagnosis: investigated group (a), control group (b). Letters a and b indicate statistically significant mean differences between the two groups *p* < 0.05, A, B, C, D, E, F, G, and H indicate statistically significant differences between days in the same group.

**Figure 5 animals-13-02134-f005:**
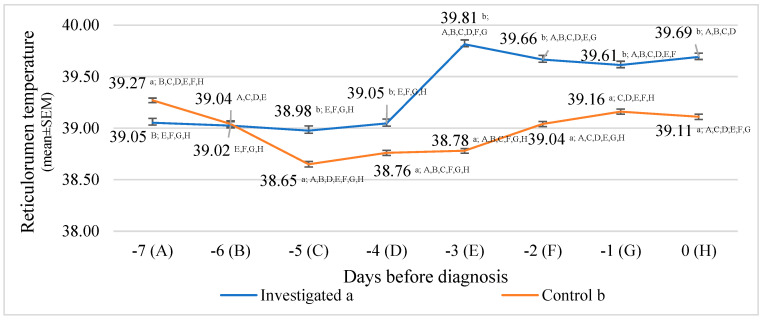
Comparison of reticulorumen temperature between days before diagnosis in two groups of cows (n = 293): investigated group (a), control group (b). Letters a and b indicate statistically significant mean differences between the two groups. A, B, C, D, E, F, G, and H indicate statistically significant differences between days in the same group.

**Figure 6 animals-13-02134-f006:**
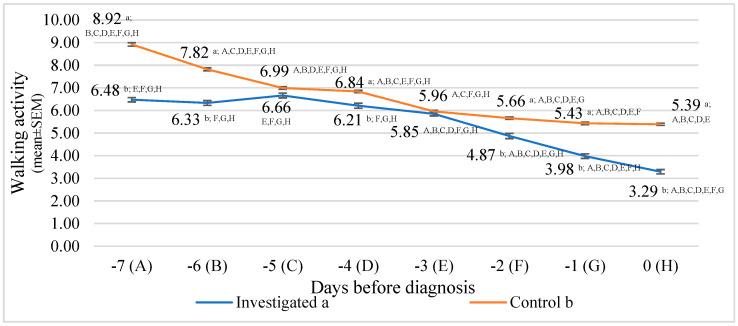
Walking activity comparison between the control and investigated groups days before diagnosis. Letters a and b indicate statistically significant mean differences between the two groups: investigated group (a), control group (b). A, B, C, D, E, F, G, and H indicate statistically significant differences between days in the same group.

**Table 1 animals-13-02134-t001:** Investigated traits means (M) and standard errors (SE) in a group of cows.

Group	pH	Temperature	Walking Activity
Investigated ^a^ (n = 25)	5.70 ± 0.009 *** ^b^	39.36 ± 0.011 * ^b^	5.47 ± 0.027 *** ^b^
Control group ^b^ (n = 25)	6.15 ± 0.038 *** ^a^	38.87 ± 0.020 * ^a^	6.62 ± 0.112 *** ^a^

Letters ^a^ and ^b^ indicate statistically significant differences between means of different groups. * *p* < 0.05, *** *p* < 0.001.

**Table 2 animals-13-02134-t002:** Correlation between investigated indicators with the days before diagnosis.

Indicator	Investigated Group (CM)	Controlled Group (Healthy)
Reticulorumen pH	−0.312 ***	0.023 **
Reticulorumen temperature	0.208 ***	−0.064 ***
Walking activity	−0.312 ***	−0.378 ***

** *p* < 0.01, *** *p* < 0.001.

**Table 3 animals-13-02134-t003:** Analysis of factors that contributed to the possibility of mastitis using a logistic regression model: 1—CM cows, 2—controlled cows.

Indicators	Classes of Explanatory Variables	B	S.E.	Wald	df	*p*-Value	OR: Odds Ratio Exp(B)	(95% CI for OR)
Reticulorumen temperature	1—investigated 2—controlled	0.231	0.010	492.357	1	<0.001	1.260	1.234–1.286
Walking activity		−0.133	0.004	1064.089	1	<0.001	0.875	0.868–0.882
Reticulorumen pH		−1.855	0.035	2745.394	1	<0.001	0.156	0.146–0.168
constant		3.021	0.471	41.186	1	<0.001	20.505	

B: unstandardized regression weight; S.E.: standard error; Wald: the test statistic for the individual predictor variable; df: degrees of freedom; *p*-value (statistically significant when *p* < 0.05); OR: odds ratio; 95% CI OR: 95% confidence interval for odds ratios.

## Data Availability

The data presented in this study are available within the article.

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
