# Peer review of "The Relationship between Reticuloruminal Temperature, Reticuloruminal pH, Cow Activity, and Clinical Mastitis in Dairy Cows"

_animals, 2023, doi:10.3390/ani13132134_

Round 1
Reviewer 1 Report (New Reviewer)
This paper describes a case-control study to compare data collected by a reticuloruminal bolus (pH, temperature, and activity) in cows in the week before being diagnosed with clinical mastitis, to those that remained healthy. Unfortunately, there are serious flaws in this study's conceptualization, execution, analysis, and reporting and I cannot recommend this paper be accepted.
The scope of this paper is confusing and appears misguided. The word “predict” is used in the title but this is not reflected in the study design or analysis. The term “biomarker” is used throughout but the collected data are not biomarkers. The focus of the introduction and discussion on automatic milking systems and zoonotic Staph. aureus infections are of little relevance to the study design.
The experimental design is poorly described and as such this is difficult to appraise. Important steps in the methodology, such as when rumen sensors were administered, are omitted. The outcome is described as clinical signs of mastitis but the approach and personnel involved in the diagnosis are not provided, this is critical information. No information is provided, or consideration given, to different grades of clinical mastitis and how this would affect the collected data which all relate to systemic changes detected by the rumen sensors. Cases are described as being clinically normal but the implication from results is that cases had increased reticuloruminal temperature and decreased pH, the latter discussed as reflecting possible SARA or reduced DMI. Therefore it seems likely that some detectable abnormalities would be present on a clinical exam. The selection of control animals is fundamental to this study design and could bias results considerably if not conducted appropriately. This important step is not described in sufficient detail. The distribution of DIM in the study population is very narrow which also questions the validity of this study population given cases were diagnosed over 6 months.
The statistical analysis is inadequate. A large part of the analysis is dedicated to analyzing the various sensor data as dependent variables, which has little bearing on the stated objective of assessing these variables for an association with clinical mastitis. The repeated measures from each animal in the week before clinical mastitis was diagnosed are not modelled correctly. Most of the results are presented as percentage changes but where raw data is provided some of the reported percentages appear incorrect. A large number of statistical tests have been conducted, with limited information on the tests being used and no consideration given to the inflated type one error with this approach. Reticuloruminal pH is categorized into four bins with no justification for this widely discouraged approach (incidentally, the cited reference appears incorrect here). Many of the figures (e.g. 4, 5, 6) present mean results without an indication of distribution/error, and include such a large number of pairwise tests that they become unreadable. In short, the analysis is inappropriate for the objectives of this paper and it is therefore difficult to assess whether the results have been appropriately interpreted, and impossible to determine the scientific value of this study.
The overall quality of English is acceptable. There are several sentences which are unclear and should be re-phrased, but I would consider this a minor issue at this stage.
Author Response
Dear Reviewer,
Authors are very thankful for the comments, which help us to improve the manuscript. All changes proposed have been included in the manuscript and highlighted in yellow and track changes.
Best Regards,
Prof. Ramunas Antanaitis
Question |
Answers |
The scope of this paper is confusing and appears misguided. The word “predict” is used in the title but this is not reflected in the study design or analysis. The term “biomarker” is used throughout but the collected data are not biomarkers.
The focus of the introduction and discussion on automatic milking systems and zoonotic Staph. aureus infections are of little relevance to the study design. |
We corrected tittle to – “The Relationship Between Reticuloruminal Temperature, Reticuloruminal pH, Cow Activity, And Clinical Mastitis In Dairy Cows” We changed “biomarker” to “parameter”
In introduction section we added information – “Because of technological advancements and growing interest in AMDs continuous measurement of reticular pH is now possible using indwelling devices (boluses) [23]” |
The experimental design is poorly described and as such this is difficult to appraise. Important steps in the methodology, such as when rumen sensors were administered, are omitted.
The outcome is described as clinical signs of mastitis but the approach and personnel involved in the diagnosis are not provided, this is critical information.
No information is provided, or consideration given, to different grades of clinical mastitis and how this would affect the collected data which all relate to systemic changes detected by the rumen sensors.
Cases are described as being clinically normal but the implication from results is that cases had increased reticuloruminal temperature and decreased pH, the latter discussed as reflecting possible SARA or reduced DMI. Therefore it seems likely that some detectable abnormalities would be present on a clinical exam. The selection of control animals is fundamental to this study design and could bias results considerably if not conducted appropriately. This important step is not described in sufficient detail.
The distribution of DIM in the study population is very narrow which also questions the validity of this study population given cases were diagnosed over 6 months. |
We added information – “The parameters of reticulorumen contents were assessed using smaXtec boluses (smaXtec animal care technology®, Graz, Austria), designed specifically for animal care. SmaXtec animal care technology® allows for the continuous presentation of data such as ruminal pH and temperature in real time. The boluses were injected into the reticulorumens using a special tool, as directed by the manufacturer. One bolus is delivered orally per cow and is retained in the reticulum due to gravity. The boluses were activated before application, cross-referenced with the cow's unique ear tag number, and a connection to the base station was established. The device's proper operation was checked, and it was calibrated in a buffer solution with a pH of 7.00 (Reagecon Buffer Solution 10702550, Reagecon, Shannon, Ireland). Following a general examination of cows in a self-locking grid, heads were manually restrained by the individual applying the bolus. Opening the mouths and applying boluses with the proper applicator (1.34-inch balling gun). Boluses were transported to the base of the tongue and ingested voluntarily by cows. Cows were then watched for two hours for any undesirable effects. The data was collected using antennas (smaXtec animal care technology®). An indwelling and wireless data-transmitting device was used to monitor RT, pH, TRR, and cow activity (smaXtec animal care GmbH, Graz, Austria). A microprocessor controlled the system. Data on pH and TRR were acquired using an A/D converter and stored on an external memory chip for further analysis and interpretation. The smaXtec messenger® computer software collected all data.”
We added information – „From 01 July to 31 December 2022, 28 clinical mastitis cases were diagnosed by a local veterinarian among the herd (n = 600) by clinical symptoms, such as abnormal milk appearance (watery, flakes, fibrin clots, etc.; mild), abnormal milk appearance with a swollen or painful quarter, abnormal milk appearance with a swollen or painful quarter, and systemic signs of illness (fever, decreased appetite, dehydration, etc.; severe). In this research, we included cows with all four infected quarters”
We added information – „From 01 July to 31 December 2022, 28 clinical mastitis cases were diagnosed by a local veterinarian among the herd (n = 600) by clinical symptoms, such as abnormal milk appearance (watery, flakes, fibrin clots, etc.; mild CM), abnormal milk appearance with a swollen or painful quarter (moderate CM), abnormal milk appearance with a swollen or painful quarter, and systemic signs of illness (fever, decreased appetite, dehydration, etc.; severe). Cows with mild and moderate CM were included in the study. In this research, we included cows with all four infected quarters”
We corrected information – „A general clinical examination revealed that no cows had clinical indications compatible with disease or other variables such as heat stress or estrus. Cows with these characteristics (n = 3) were excluded from the research. Cows classified as control did not have mastitis during their current lactation. This group’s total number of cows was 25 (195.65 ± 2.5 days in milk). If the cow met the enrollment criteria, local veterinarian aseptically collected a single milk sample from the affected quarter”
The distribution of lactation days in this study was as reported in the manuscript We added information – “This group’s total number of cows was 28 (187.5 ± 5 days in milk)”
|
The statistical analysis is inadequate. A large part of the analysis is dedicated to analyzing the various sensor data as dependent variables, which has little bearing on the stated objective of assessing these variables for an association with clinical mastitis. The repeated measures from each animal in the week before clinical mastitis was diagnosed are not modelled correctly. Most of the results are presented as percentage changes but where raw data is provided some of the reported percentages appear incorrect. A large number of statistical tests have been conducted, with limited information on the tests being used and no consideration given to the inflated type one error with this approach. Reticuloruminal pH is categorized into four bins with no justification for this widely discouraged approach (incidentally, the cited reference appears incorrect here). Many of the figures (e.g. 4, 5, 6) present mean results without an indication of distribution/error, and include such a large number of pairwise tests that they become unreadable. In short, the analysis is inappropriate for the objectives of this paper and it is therefore difficult to assess whether the results have been appropriately interpreted, and impossible to determine the scientific value of this study. |
Since we studied reticulorumen pH, reticulorumentemperature, walking activity in cows with clinical mastitis and controlled/ healthy cows groups, the analysed data and results showed a relationship and a change comparing the results during the days before diagnosis and differences between cows with clinical mastitis and controlled/healthy cows. Also we used correlation analysis between days before diagnosis and investigated indicators, for finding out whether a relationship exists between variables. The significance in figures was corrected to a clearer version. All the test used for the data analysis is mentioned in the material and methods part. Also we used logistic regression analysis for analysis of probability of mastitis risk model ( investigated group – cows with clinical mastitis, control – healthy cows). |
The overall quality of English is acceptable. There are several sentences which are unclear and should be re-phrased, but I would consider this a minor issue at this stage. |
English was corrected by the MDPI English editing service.
|
Reviewer 2 Report (Previous Reviewer 2)
The methods of microbiological experiments should be described in more detail.
line 163 not Staph. aureus but S. aureus (in italics of course)
Author Response
Dear Reviewer,
Authors are very thankful for the comments, which help us to improve the manuscript. All changes proposed have been included in the manuscript and highlighted in yellow and track changes.
Best Regards,
Prof. Ramunas Antanaitis
Question |
Answers |
The methods of microbiological experiments should be described in more detail. |
We added information – “All cows’ (n=25) milk samples were collected for microbiological investigation. Using the techniques indicated in the National Mastitis Council guidelines [34], detailed laboratory protocols are reported elsewhere [35]…” |
line 163 not Staph. aureus but S. aureus (in italics of course) |
Corrected |
Reviewer 3 Report (New Reviewer)
Dear Authors,
The manuscript presents the measurement data on healthy cows and cows with mastitis, specifically data on reticulorumen temperature, pH and walking activity. I have a few minor points:
1) Figure 6, y-axis has a spelling mistake in "Walking". Figure 4 x-axis has spelling mistake in "before". The fonts of x-y axis of Fig 6 could be same as those of Figure 4 and 5. Figure 1 y-axis will be "number of cows". Fig 3-6 should say "Control group" and not "controlled group"
2) Some introduction of the smaXtec system and how it works could be provided in Section on Measurements (2.4.1).
3) In Figure 1, how are the 4 classes created and grouped? Is there a need or value for this grouping?
4) In Fig 4-6, the standard deviation is missing in the data. How is the walking activity defined or measured, and was it validated by a human subject?
5) A recent article mentions several parameters that could be used in the measurements. Your discussion section can discuss some other possible parameters in addition to pH, temperature, and walking activity. Animals 2021, 11(9), 2665; https://doi.org/10.3390/ani11092665
Author Response
Dear Reviewer,
Authors are very thankful for the comments, which help us to improve the manuscript. All changes proposed have been included in the manuscript and highlighted in yellow and track changes.
Best Regards,
Prof. Ramunas Antanaitis
Question |
Answers |
1) Figure 6, y-axis has a spelling mistake in "Walking". Figure 4 x-axis has spelling mistake in "before". The fonts of x-y axis of Fig 6 could be same as those of Figure 4 and 5. Figure 1 y-axis will be "number of cows". Fig 3-6 should say "Control group" and not "controlled group" |
Corrected. |
2) Some introduction of the smaXtec system and how it works could be provided in Section on Measurements (2.4.1). |
We added information – “The parameters of reticulorumen contents were assessed using smaXtec boluses (smaXtec animal care technology®, Graz, Austria), designed specifically for animal care. SmaXtec animal care technology® allows for the continuous presentation of data such as ruminal pH and temperature in real time. The boluses were injected into the reticulorumens using a special tool, as directed by the manufacturer. One bolus is delivered orally per cow and is retained in the reticulum due to gravity. The boluses were activated before application, cross-referenced with the cow's unique ear tag number, and a connection to the base station was established. The device's proper operation was checked, and it was calibrated in a buffer solution with a pH of 7.00 (Reagecon Buffer Solution 10702550, Reagecon, Shannon, Ireland). Following a general examination of cows in a self-locking grid, heads were manually restrained by the individual applying the bolus. Opening the mouths and applying boluses with the proper applicator (1.34-inch balling gun). Boluses were transported to the base of the tongue and ingested voluntarily by cows. Cows were then watched for two hours for any undesirable effects. The data was collected using antennas (smaXtec animal care technology®). An indwelling and wireless data-transmitting device was used to monitor RT, pH, TRR, and cow activity (smaXtec animal care GmbH, Graz, Austria). A microprocessor controlled the system. Data on pH and TRR were acquired using an A/D converter and stored on an external memory chip for further analysis and interpretation. The smaXtec messenger® computer software collected all data.”
|
4) In Fig 4-6, the standard deviation is missing in the data. How is the walking activity defined or measured, and was it validated by a human subject? |
Corrected.
We added information- “The parameters of reticulorumen contents and cow walking activity were assessed using smaXtec boluses (smaXtec animal care technology®, Graz, Austria)” |
5) A recent article mentions several parameters that could be used in the measurements. Your discussion section can discuss some other possible parameters in addition to pH, temperature, and walking activity. Animals 2021, 11(9), 2665; https://doi.org/10.3390/ani11092665 |
We added in discussion section – “Relevant behavioral health indicators are described in order to integrate machine learning algorithms and decision support systems to detect animal lameness, lethargy, pain, injury, and distress. The roadmap for technology adoption is also highlighted, as are the difficulties and next steps. The technology has the potential to result in more effective farm animal management, a more focused emphasis on sick animals, medical expense reductions, and fewer antibiotic use [56]“
|
Round 2
Reviewer 1 Report (New Reviewer)
Thank you for addressing some of the previous comments. There are still some important details that need to be included in the methods and I think further analysis is required. I suggest major revisions are made prior to publication.
I am still unclear whether every cow in the herd had a rumen bolus administered or if some cows were pre-selected for monitoring. If all cows on the herd had these boluses then please explain this, if it was only a subset of cows that had boluses then it should be described how these cows were selected.
Thank you for including details of how mastitis cases were diagnosed, however further details are still needed. Please clarify whether the local veterinarian assessed every cow every day, or just confirmed a diagnosis made by the farm staff. It is also important to understand how controls were defined – please describe what examinations were conducted on control animals and when, i.e. how it was determined that they were healthy.
It is important to present and discuss how many cows had mild, moderate or serve mastitis cases. The interpretation of the results of this study would be very different if all cases were mild or all cases were severe.
As per my previous comment, the selection of controls is a fundamental part of this study design. Please describe this in detail.
Presenting all results as percentage changes without absolute figures can be misleading, please provide the absolute changes on each occasion in the text. This can be a particular problem when interpreting average percentage changes without accounting for the range within each group. For example, the conclusion (L454 to 460) needs to take this into account and while differences may be seen on average, they may not be consistent enough to be used for early diagnosis. All interpretation needs to consider this.
I cannot follow how some of the percentage changes have been calculated e.g. L283 “93.22% more cows in the first class of reticulorumen pH than in the fourth class” but there appear to be 26 animals in the first class and 4 animals in the fourth class (reading from Figure 1 as instructed in the text (L241)). Also, a different % is reported when comparing the first class (n=26) to the second class (n=10) than the first class (n=26) to the third class (n=10).
Throughout, please be consistent with decimal places as periods or commas, refer to journal guidelines.
Specific points
L224 – I am still unclear why rumen pH has been categorized into four groups and not analysed as a continuous variable, statistically this is bad practice so needs a clear explanation in the text. It is referenced to “our previous publication” but the citation appears incorrect – 35, Lago, A.; Godden, S.M.; Bey, R.; Ruegg, P.L.; Leslie, K. The Selective Treatment of Clinical Mastitis Based on On-Farm Culture Results: I. Effects on Antibiotic Use, Milk Withholding Time, and Short-Term Clinical and Bacteriological Outcomes. J. Dairy Sci. 2011, 94, 4441–4456, doi:10.3168/jds.2010-4046.
L238 to 241 - Chi-squared tests on the differences between rumen pH groups do not make sense to me – what hypothesis are you testing? i.e. “p = 0.001” therefore rejects the null hypothesis of what? Same for L249 to 251.
L247 “Data analysis of reticulorumen temperature revealed a statistically significant relationship between classes of reticulorumen pH.” Please clearly state what test has been used and what the result was - just describing “statistical significance” is not sufficient.
Figure 3 – Please make it clear when the 24 hour window shown in this figure occurred. If this is the average over the whole week before diagnosis then please make this clear. Please provide some indication of the distribution or error of the mean i.e. by adding a shaded area around the line.
L292 – You need to account for repeated measurements for each cow in this linear regression model. The number of days before diagnosis does not have a linear relationship with rumen pH i.e. there is no difference between days -7 and day -4, therefore you need to use a statistical approach to account for this non-additivity. Same for rumen temperature and the number of days before diagnosis (L313).
Figure 4 – Please describe what the error bars indicate and add these for the control group too.
Table 2 (the first one) – This table appears to have been corrupted in formatting, please resubmit.
L349 – Logistic regression does not model risk, please correct the interpretation.
Table 2 (the second one) – Please make it clear which measurements were modelled with logistic regression, and whether this logistic regression model accounts for repeated measurements. If repeated measurements were not modelled then please re-analyse with a random effect term and describe the effect of the grouping factor (cow) in the model.
Author Response
Dear Reviewer,
Authors are very thankful for the comments, which help us to improve the manuscript. All changes proposed have been included in the manuscript and highlighted in yellow and track changes.
Best Regards,
Prof. Ramunas Antanaitis
Question |
Answers |
|||||||||||||||||||||||||||||||||||||||||||||||||||||||||||||||||||||||||||||||||||||||||||||||||||||||||||||||||||||||||||
I am still unclear whether every cow in the herd had a rumen bolus administered or if some cows were pre-selected for monitoring. If all cows on the herd had these boluses then please explain this, if it was only a subset of cows that had boluses then it should be described how these cows were selected. |
We added information – “The reticulorumen parameters and walking activity of all cows (n = 600) were assessed using smaXtec boluses (smaXtec animal care technology®, Graz, Austria), designed specifically for animal care” |
|||||||||||||||||||||||||||||||||||||||||||||||||||||||||||||||||||||||||||||||||||||||||||||||||||||||||||||||||||||||||||
Thank you for including details of how mastitis cases were diagnosed, however further details are still needed. Please clarify whether the local veterinarian assessed every cow every day, or just confirmed a diagnosis made by the farm staff. It is also important to understand how controls were defined – please describe what examinations were conducted on control animals and when, i.e. how it was determined that they were healthy. |
We corrected to – “From 01 July to 31 December 2022, 28 clinical mastitis cases from herd of 600 cows were identified by the farm staff in the milking parlor and confirmed by local veterinarian based on clinical indications of mastitis by clinical symptoms, such as abnormal milk appearance (watery, flakes, fibrin clots, etc.; mild CM), abnormal milk appearance with a swollen or painful quarter (moderate CM), abnormal milk appearance with a swollen or painful quarter, and systemic signs of illness (fever, decreased appetite, dehydration, etc.; severe)”
We corrected to – “Twenty-five cows (second and more lactation numbers and 193.65 ± 2.4 days in milk) were categorized as clinically healthy with an SCC of <200,000 cells/mL and no clinical signs of disease, such as abnormal milk appearance (watery, flakes, fibrin clots, etc.; mild CM), abnormal milk appearance with a swollen or painful quarter (moderate CM), abnormal milk appearance with a swollen or painful quarter, and systemic signs of illness (fever, decreased appetite, dehydration, etc.; severe)”
|
|||||||||||||||||||||||||||||||||||||||||||||||||||||||||||||||||||||||||||||||||||||||||||||||||||||||||||||||||||||||||||
It is important to present and discuss how many cows had mild, moderate or serve mastitis cases. The interpretation of the results of this study would be very different if all cases were mild or all cases were severe. As per my previous comment, the selection of controls is a fundamental part of this study design. Please describe this in detail. |
In methods section we added information – “Cows with mild and moderate CM were included in the study”. Severe mastitis cases were not included in this study.
|
|||||||||||||||||||||||||||||||||||||||||||||||||||||||||||||||||||||||||||||||||||||||||||||||||||||||||||||||||||||||||||
Presenting all results as percentage changes without absolute figures can be misleading, please provide the absolute changes on each occasion in the text. This can be a particular problem when interpreting average percentage changes without accounting for the range within each group. For example, the conclusion (L454 to 460) needs to take this into account and while differences may be seen on average, they may not be consistent enough to be used for early diagnosis. All interpretation needs to consider this. |
We think that presenting all results as percentages without absolute numbers can be misleading. Provide absolute changes in the text each time. This can be a particular problem when interpreting mean percentage changes without considering the range within each group. |
|||||||||||||||||||||||||||||||||||||||||||||||||||||||||||||||||||||||||||||||||||||||||||||||||||||||||||||||||||||||||||
I cannot follow how some of the percentage changes have been calculated e.g. L283 “93.22% more cows in the first class of reticulorumen pH than in the fourth class” but there appear to be 26 animals in the first class and 4 animals in the fourth class (reading from Figure 1 as instructed in the text (L241)). Also, a different % is reported when comparing the first class (n=26) to the second class (n=10) than the first class (n=26) to the third class (n=10). |
Corrected |
|||||||||||||||||||||||||||||||||||||||||||||||||||||||||||||||||||||||||||||||||||||||||||||||||||||||||||||||||||||||||||
Throughout, please be consistent with decimal places as periods or commas, refer to journal guidelines. |
Corrected |
|||||||||||||||||||||||||||||||||||||||||||||||||||||||||||||||||||||||||||||||||||||||||||||||||||||||||||||||||||||||||||
Specific points |
|
|||||||||||||||||||||||||||||||||||||||||||||||||||||||||||||||||||||||||||||||||||||||||||||||||||||||||||||||||||||||||||
L224 – I am still unclear why rumen pH has been categorized into four groups and not analysed as a continuous variable, statistically this is bad practice so needs a clear explanation in the text. It is referenced to “our previous publication” but the citation appears incorrect – 35, Lago, A.; Godden, S.M.; Bey, R.; Ruegg, P.L.; Leslie, K. The Selective Treatment of Clinical Mastitis Based on On-Farm Culture Results: I. Effects on Antibiotic Use, Milk Withholding Time, and Short-Term Clinical and Bacteriological Outcomes. J. Dairy Sci. 2011, 94, 4441–4456, doi:10.3168/jds.2010-4046. |
The r.pH was analyzed as categorized variable for association in prediction of dairy cow health status and for relationship/ changed of temperature, walking activity with the change of pH class.
We corrected to – “The experimental animals were divided into four classes based on the reticulorumen pH assay: first class <6.22; second class 6.22–6.42; third class 6.42–6.62; fourth class >6.62. Classes were assigned according to our previous publication [40]”
|
|||||||||||||||||||||||||||||||||||||||||||||||||||||||||||||||||||||||||||||||||||||||||||||||||||||||||||||||||||||||||||
L238 to 241 - Chi-squared tests on the differences between rumen pH groups do not make sense to me – what hypothesis are you testing? i.e. “p = 0.001” therefore rejects the null hypothesis of what? Same for L249 to 251. |
Answer: Hypothesis: the probability of a cow falling into any of the 4 classes is equal.
|
|||||||||||||||||||||||||||||||||||||||||||||||||||||||||||||||||||||||||||||||||||||||||||||||||||||||||||||||||||||||||||
L247 “Data analysis of reticulorumen temperature revealed a statistically significant relationship between classes of reticulorumen pH.” Please clearly state what test has been used and what the result was - just describing “statistical significance” is not sufficient. |
Repeated measurements – Paired sample test was used for comparing means.
Corrected to
Data analysis of the differences in arithmetic means of reticulorumen temperature revealed a statistically significant relationship between classes of reticulorumen pH.
|
|||||||||||||||||||||||||||||||||||||||||||||||||||||||||||||||||||||||||||||||||||||||||||||||||||||||||||||||||||||||||||
Figure 3 – Please make it clear when the 24 hour window shown in this figure occurred. If this is the average over the whole week before diagnosis then please make this clear. Please provide some indication of the distribution or error of the mean i.e. by adding a shaded area around the line. |
Corrected The mean of reticulorumen pH over 24 h in a group of cows. So we can see the changes per 24 hours. Corrected.
|
|||||||||||||||||||||||||||||||||||||||||||||||||||||||||||||||||||||||||||||||||||||||||||||||||||||||||||||||||||||||||||
L292 – You need to account for repeated measurements for each cow in this linear regression model. The number of days before diagnosis does not have a linear relationship with rumen pH i.e. there is no difference between days -7 and day -4, therefore you need to use a statistical approach to account for this non-additivity. Same for rumen temperature and the number of days before diagnosis (L313) |
Test for a linear trend in the investigated traits means, based on the values for the factor levels: days before diagnosis,(results in a table below); after that it was decided to show the relation of DAYS change with investigated trait, using linear regression analysis. The results show that the variation in the means can be described in a linear line…. Look at Sig. column.
Remark: P< 0.001 considered significant; the line can reliably describe the change in means.
|
|||||||||||||||||||||||||||||||||||||||||||||||||||||||||||||||||||||||||||||||||||||||||||||||||||||||||||||||||||||||||||
Figure 4 – Please describe what the error bars indicate and add these for the control group too. |
Error bars indicate standard error of mean. The figure was corrected, lines changed to width 1 pt, for better view of standard errors of means.
|
|||||||||||||||||||||||||||||||||||||||||||||||||||||||||||||||||||||||||||||||||||||||||||||||||||||||||||||||||||||||||||
Table 2 (the first one) – This table appears to have been corrupted in formatting, please resubmit. |
Design/format of table 2 was corrected. |
|||||||||||||||||||||||||||||||||||||||||||||||||||||||||||||||||||||||||||||||||||||||||||||||||||||||||||||||||||||||||||
L349 – Logistic regression does not model risk, please correct the interpretation. |
to predict the chance of mastitis - odds ratio. We used a logistic regression model to determine associations of factors that contributed to the possibility of mastitis. |
|||||||||||||||||||||||||||||||||||||||||||||||||||||||||||||||||||||||||||||||||||||||||||||||||||||||||||||||||||||||||||
Table 2 (the second one) – Please make it clear which measurements were modelled with logistic regression, and whether this logistic regression model accounts for repeated measurements. If repeated measurements were not modelled then please re-analyse with a random effect term and describe the effect of the grouping factor (cow) in the model.
|
The model of logistic regression was used from means (not direct repeated measurement of each cow), because we have tried with repeated measures and a random effect of cow, but computer resources do not allow calculation (it showed running GENLINMIXED… more than 24 hours and we couldn’t receive the calculations. We had by force to close the program. So in the manuscript is presented Binary logistic regression without random effect of cow.
|
This manuscript is a resubmission of an earlier submission. The following is a list of the peer review reports and author responses from that submission.
Round 1
Reviewer 1 Report
Dear Author,
I hope this letter finds you well. I have had the opportunity to review your manuscript entitled, which I found to be an interesting and thought-provoking piece of work. However, I have some concerns regarding the statistical analysis and presentation of data in the manuscript.
In my assessment, I noticed that there were several instances of bias in the statistical analysis and interpretation of the results. I believe that this undermines the credibility of your research and could potentially lead to erroneous conclusions.
Specific,
Line 39 The lowest walking activity in the CM group was detected 0 days before diagnosis, 50.60% lower than on the fifth day before diagnosis. The lowest walking activity was detected 0 days before diagnosis, 39.57% lower than on the seventh day before diagnosis." - It is redundant to state that the lowest walking activity was detected 0 days before diagnosis twice. This should be corrected.
Line 94- 95 IMI due to the feverish state caused by the immune response to infections 94 [2122], the reference numbers should be separated by a comma
Line 118 In 2.1, it would be helpful to specify who provided the ethical approval (e.g., an institutional review board, an animal care committee). Also, it would be useful to state how the ethical approval was obtained (e.g., review of a written protocol, in-person review of the experimental setup).
Line 124 Define DM
Line 139 – 146 paragraph space is different from the other text.
Line 166 167 we could monitor real-time data such as pH, reticulorumen content (TRR) temperature, and cow activity [3334] the reference numbers should be separated by a comma
Line 208 “The reticulorumen temperature 208 decreased by about 0.029 (y = -0.029x + 39.02; R² = 0.0857). Temperature units must be referred
Line 229 table 1. Controlled (should be control group) are 75 but in line 160 “Twenty-five cows (second and more lactation numbers and 193.65 ± 2.4 days in milk) were categorized as clinically healthy with an SCC of <200,000 cells/mL and no clinical signs of disease.
Line 232 These are changes in specific day or they are the mean values of every 24 h?
Line 242 days before diagnosis should be deleted or should be days before diagnosis in CM group
Line 244 What do you mean in a group of cow? Who was that group?
Line 244 to 259 this information is not scientifically useful, the results of repeated measurements should be presented
Line 266 – 268. what is the significance of this comparison and how can it be scientifically explained?
Line 285 what is the significance of this comparison and how can it be scientifically explained?
Line 309 A non-significant with p <0.05?
Line 313 Boluses need to be explained!
Major issues:
In statistical analysis "Repeated measures analysis of variance (ANOVA) was used for comparing means across the investigated variables based on observations from -7 to 0 days before diagnosis." Where are the results of tha analysis?
Comparing highlest or lowest mean values does not provide any useful scientific result especially in control group!
More over the analysis should be preformed as logistic regression and then according to the results to selected linear as more appropriate..
In line 189 A backward stepwise logistic model is referred.....where are these results?

Reviewer 2 Report
How many quarters out of 25 cows were affected by clinical mastitis?
Lines 49-75 of the Introduction concern S. aureus but in chapter Materials and Method, only two sentences were devoted to microbiological examinations (All cow's milk samples (n=25) were collected for microbiological tests. Microbiological tests for Streptococcus spp. and S. aureus, and for susceptibility to amoxicillin and clavulanic acid).
How many milk samples have been microbiologically examined?
What microbiological media were used in microbiological diagnostics?
Have CF (clumping factor) test and other tests been performed?
How many cows and quarters were infected with S. aureus and how many with streptococci?
Has S. aureus been tested for MRSA?
Have PCR tests been performed?
Were milk samples tested for yeast and Prototheca?
line 148 Streptococcus – italics
In the Introduction, articles on S. aureus transmission between humans and cows should be cited (e.g. Sato et al.: Closely related methicillin-resistant Staphylococcus aureus isolates from retail meat, cows with mastitis, and humans in Japan. PLoS One 2017; Magro et al.: Methicillin-resistant Staphylococcus aureus CC22-MRSA-IV as an agent of dairy cow intramammary infections. Vet. Microbiol. 2018; Krukowski et al.: The first outbreak of methicillin-resistant Staphylococcus aureus in dairy cattle in Poland with evidence of on-farm and intrahousehold transmission. J. Dairy Sci. 2020)